# Rice Porridge Containing Welsh Onion Root Water Extract Alleviates Osteoarthritis-Related Pain Behaviors, Glucose Levels, and Bone Metabolism in Osteoarthritis-Induced Ovariectomized Rats

**DOI:** 10.3390/nu11071503

**Published:** 2019-06-30

**Authors:** Hye Jeong Yang, Min Jung Kim, Jing Yi Qiu, Ting Zhang, Xuangao Wu, Dai-Ja Jang, Sunmin Park

**Affiliations:** 1Food Functional Research Division, Korean Food Research Institutes, Wanjoo 55365, Korea; 2Department of Food and Nutrition, Obesity/Diabetes Center, Hoseo University, 165 Sechul-Ri, BaeBang-Yup, Asan-Si, ChungNam-Do 31499, Korea

**Keywords:** Welsh onion root, rice porridge, estrogen-deficiency, osteoarthritis, inflammation, pain

## Abstract

Rice porridge containing *Allium fistulosum* (Welsh onion) root water extract (RAFR) has anti-inflammatory bioactive compounds. We examined whether the long-term administration of rice porridge with RAFR would prevent or delay the progression of osteoarthritis and menopausal symptoms in estrogen-deficient animals by ovariectomy. The rats consumed 40% fat energy diets containing 250 mg RAFR (rice: *Allium fistulosum* root = 13:1)/kg body weight (bw) (OVX-OA-RAFR-Low), 750 mg RAFR/kg bw (OVX-OA-RAFR-High) and 750 mg starch and protein/kg bw(OVX), respectively. After consuming the assigned diets for eight weeks, monoiodoacetate (OVX-OA) or saline (OVX) were injected into the knee joints of the rats for an additional three weeks. Sham rats were administered saline injections (normal-control). OVX-OA-RAFR improved oral glucose tolerance and also protected against decreases in bone mineral density and lean body mass in the legs and increases in fat mass in the abdomen, compared to the OVX and OVX-OA. OVX-OA-RAFR improved swelling and limping scores, normalized weight distribution between the osteoarthritic and normal limbs, and increased maximum running speeds compared to the OVX-OA. The OVX-OA deteriorated the articular cartilage by reducing the articular matrix and bone loss in the knee joint and it prevented knee joint deterioration when compared to the OVX. The improvement in osteoarthritis symptoms in OVX-OA-RAFR decreased the mRNA expression of matrix metallo-proteinase-1 and matrix metalloproteinase-13, tumor necrosis factor-α, and interleukin-1β and interleukin-6 in the articular cartilage compared to OVX-OA rats. In conclusions, RAFR is effective in treating osteoarthritis symptoms and it may be used for a therapeutic agent in osteoarthritis-induced menopausal women.

## 1. Introduction

Osteoarthritis (OA) is a disease of articular joints induced by increased inflammation. The prevalence of this disease increases with age, and the majority of individuals over the age of 65 are affected by OA [1]. OA is a primary cause of pain contributing to the immobility of elderly people and is also a leading cause of disability globally [2]. A previous meta-analysis demonstrated that the overall prevalence of radiographic knee OA is about 35% and that this prevalence increases with age in the USA. Among Korean people, 9.3% of male and 28.5% of female participants were diagnosed with symptomatic OA according to survey criteria in the Fifth Korean National Health and Nutrition Examination Survey (2010–2012) [3]. In China, the prevalence of symptomatic knee OA is about 8.1% and is much higher in women (10.3%) than in men (5.7%) [4]. 

Cartilage loss is considered to be the pathological aspect of OA. However, the disease extends throughout the whole joint, including the bone and soft tissues such as the synovium, menisci, and ligaments [5]. The main symptoms of OA are a pain, stiffness, and loss of function. The pathophysiology remains unknown although, in general, it is initiated with injury and excessive usage, which induces inflammation causing the removal of articular cartilage. OA decreases quality of life due to pain and the limitation of physical activity [6]. The risk factors for OA at a personal level include age, gender, weight, skeletal muscle mass, menopausal status, genetics, and diet. The risk factors for OA at the joint level, including injury, misalignment, and abnormal loading of the joints. Personal and joint level risk factors interact in a complex manner to aggregate OA symptoms [6]. In particular, menopausal women are highly susceptible to OA incidence. 

Estrogen deficiency exacerbates energy, glucose, lipid, and bone metabolism. This deficiency also elevates low-grade systematic inflammation with increased levels of pro-inflammatory cytokines, such as interleukin-1 (IL-1) and tumor necrosis factor-α (TNF-α) [7,8]. Menopausal women are susceptible to OA, as this disease interacts with other metabolic diseases and their symptoms are exacerbated by one another.

These diseases have common pathogenic mechanisms, such as low-grade inflammation and oxidative stress. However, beyond this common etiology, metabolic diseases have direct systemic action on the joints. In addition to the impact of weight, obesity-associated inflammation is associated with OA severity and may modulate OA progression in mouse models. Increased insulin resistance may participate in joint catabolism. Finally, the gut microbiota is also modulated by insulin resistance and inflammation to influence metabolic syndrome and OA [9,10]. 

Medicines available to treat OA are currently limited in their effectiveness. Pharmaceuticals currently used to treat OA primarily focus on reducing pain and inflammation, such as non-steroidal anti-inflammatory drugs (NSAIDs) [11]. However, NSAIDs are reported to have adverse effects. Glucosamine and chondroitin sulfate are mainly used as functional foods [11]; however, their effectiveness remains controversial. The roots of *Allium fistulosum* (Welsh onion) root (AFR) contain alliin, allicin and diallyl disulfide and have been reported to reduce body fat mass [12]. Allicin has anti-inflammatory and immune-modulatory properties due to its augmentation of the extracellular signal-regulated kinases (ERK)1/2 signaling pathway [13,14]. Gluten may elevate inflammation levels if allergies are present. Rice does not contain gluten, and rice porridge is readily available to the elderly and easy to consume. Therefore, porridge containing AFR could be a beneficial meal for treating OA.

In this study, we hypothesize that the long-term administration of rice porridge containing *Allium fistulosum* root water extract prevents or delays the progression of OA and menopausal symptoms in estrogen-deficient animals. This hypothesis was examined in ovariectomized (OVX) rats with OA induced via the intra-articular injection of monoiodoacetate (MIA). The action mechanism was also explored. MIA intra-articular injection into the knees is reported as a useful and relevant pre-clinical model of OA pain [15]. 

## 2. Materials and Methods

### 2.1. Rice Porridge Containing Allium Fistulosum Root Water Extracts

AFR was extracted with water at 90 °C for 2 h, and the supernatant was filtered using a disposable CA syringe filter unit (pore size; 0.22 µm, diameter; 25 mm, Futecs Co., Ltd., Daejeon, Korea) after centrifuging for 10 min at 4000× *g*. The supernatant was freeze-dried and stored for further use in the animal study. 

Rice porridge was prepared with rice and water by boiling for 2 h followed by freeze-drying. The test meal was prepared by mixing freeze-dried water extracts of AFR and freeze-dried rice porridge (1:13, *w*/*w*). 

### 2.2. Analysis of Index Compounds in RAFR by LC-MS/MS Analysis

The analyses were performed using an Acquity UPLC system (Waters, Milford, MA, USA) equipped with an Acquity UPLC BEH C18 column (2.1 mm × 100 mm, 1.7 µm). The mobile phase included 0.1% formic acid aqueous solution (Solvent A) and 0.1% formic acid in acetonitrile (Solvent B). A gradient elution program was followed: 0–6 min, 100–70% solvent A; 6–9 min, 70–0% solvent A; 9–9.5 min, 0–0 % solvent A; 9.5–10 min, 100% solvent A. The flow rate was set at 0.6 mL/min and the column temperature was maintained at 30 °C. The auto-sampler was conditioned at 10 °C and the injection volume was 5 μL. Mass spectrometric analyses were undertaken using a Waters Xevo TQ triple-quadrupole mass spectrometer equipped with electrospray ionization (ESI) mode. The ESI source was operated by switching between positive and negative ion modes with multiple reaction monitoring modes. Quantification was performed using the positive mode of *m/z* 147.2→ 84 for lysine, *m/z* 177.9 → 88 for alliin, *m/z* 162 → 73 for s-allyl-L-cysteine, and *m/z* 291 → 145 for γ-glutamyl-(s)-allyl-cysteine and the negative mode of *m/z* 192.9 → 134 for ferulic acid and *m/z* 301.4 → 151.1 for quercetin. The detector was operated at a cone voltage of 25 V and a capillary voltage of 3.5 kV. The source temperature was set at 150 °C, the desolvation flow was set at 800 L/h, and the desolvation gas temperature was set at 400 °C.

### 2.3. Ovariectomy or Sham-Operation

Eight-week-old female Sprague–Dawley rats (weighing 233 ± 19 g) were housed individually in stainless steel cages in a controlled environment (23 °C with a 12 h light/dark cycle). Fifty Sprague–Dawley rats purchased from DBL (Yeumsung-Kun, Korea) were acclimated in our animal facility for one week. Forty rats underwent an OVX, and ten rats had sham operations under anesthesia via the subcutaneous injection of a mixture of ketamine and xylazine (100 and 10 mg/kg body weight, respectively). A mid-ventral incision was made, and each ovary was separated after the ligation of the most proximal portion of the oviduct [16]. Both ovaries were removed with scissors. In the sham operations, rats underwent the same procedure as the OVX rats, but no ligation and removal of the ovaries were conducted. Thus, sham rats underwent the same surgical procedures as the OVX rats but retained their normal ovary functions. OVX rats were randomized into four groups, and the rats that underwent a sham operation were assigned to the Sham group. The surgery and care of the animals were conducted in accordance with the guidelines of the NIH Guide for the Care and Use of Laboratory Animals, and the International Association for the Study of Pain. The research procedures for the animal study were approved by the Animal Care and Use Committee of Hoseo University, Korea (HSIACUC-18-100).

### 2.4. Diet Preparation 

The diets were high in fat to exacerbate the progression of OA and menopausal symptoms in comparison to a low-fat diet [17,18]. The diet consisted of 45% energy (En%) from carbohydrates, 15 En% from protein and 40 En% from fats. This high-fat diet is a semi-purified modified AIN-93 formulation for experimental animals [19]. The major carbohydrate, protein, and fat sources are starch plus sugar, casein (milk protein), and lard (CJ Co., Seoul, Korea), respectively. The high-fat diet was supplemented with rice porridge containing AFR (0.23% rice + 0.018% AFR) for the OVX-OA-RAFR-L group, 0.69% rice + 0.053% AFR for the OVX-OA-RAFR-H group, and 0.053% cellulose for the control and Sham (normal-control) groups. For the control groups (OVX-OA, OVX, and Sham), the nutrient composition of rice was made equivalent to the OVX-OA-RAFR-H diet with rice protein, starch, and cellulose. Dried rice porridge and ARF powder were homogeneously mixed with a vitamin and mineral mixture and sugar. The dietary components and the mixture were sieved to remove lumps. This mixture was then mixed with the appropriate amounts of starch, casein, and lard and was sieved again. The resulting mixture was then stored at 4 °C, and the food supply was replaced every two days. The amount of the supplement administered (dosage) was calculated based on the food intake. 

### 2.5. Experimental Design

The forty OVX rats were randomly assigned to the following four groups: (1) MIA injection into the knee joint and fed a high-fat diet containing 0.23% rice + 0.018% AFR diet (OVX-OA-RAFR-L) to consume about 250 mg/kg body weight. (2) MIA injection into the knee joint and fed a high-fat diet containing 0.69% rice + 0.053% AFR diet (OVX-OA-RAFR-H) to consume about 750 mg/kg bw. (3) MIA injection into the knee joint and fed a high-fat diet containing 0.053% cellulose (OA). (4) Saline injection into the knee joint and fed a high-fat diet containing 0.053% cellulose (OVX). Ten sham rats had the same diet as the control and a saline injection (Sham; normal-control). After the OVX or sham operation, the rats were given free access to water and their assigned diet. Each experimental group consisted of ten rats.

### 2.6. MIA-Induced OA Animal Models 

After eight weeks of consuming their assigned diets, all OVX rats were anesthetized via the intramuscular injection of a ketamine and xylazine mixture (100 and 10 mg, respectively). The rats received a single intra-articular injection of MIA (4 mg/50 μL saline; Sigma Co., St. Louis, MO, USA) through the patellar ligament of the right knee using a 26-gauge needle [18]. Sham rats received a single intra-articular injection of saline into the right knee as the normal-control (Sham group). The left knees of all rats were administered a saline injection. After the MIA or saline injection, assigned diets were provided for an additional three weeks. The behavior and edema of the knee joints were then carefully observed on days 3, 7, 14, and 21. 

### 2.7. Tail Skin Temperature Measurements 

The tail skin temperature was measured using an infrared thermometer (BIO-152-IRB, Bioseb, Chaville, France) designed for small rodents at weeks eight and eleven of the experimental periods during the sleep cycle. Three measurements were taken every 10 min, and the average for each animal was considered as a single data point. 

### 2.8. Bone Mineral Density (BMD) Measurement 

A dual-energy X-ray Absorptiometer (DEXA; Norland pDEXA Sabre; Norland Medical Systems Inc., Fort Atkinson, WI, USA) was calibrated with a phantom before use. After anesthetization, the rats were laid in a prone position with posterior legs with 90° flexion of the hip, knee, and ankle. Upon the completion of scanning, BMD was determined in the right femur and knee using the DEXA instrument equipped with the appropriate software for the assessment of bone density in small animals [18]. Similarly, abdominal fat and lean body mass (LBM) was measured by DEXA.

### 2.9. Progression of OA and Pain-Related Behavior Evaluation 

At 3, 7, 14, and 21 days after MIA injection, the diameters of the knees were measured every week using digital calipers (Mitotoyo, Kawasaki-shi, Japan). All rats were carefully evaluated for knee joint swelling and walking patterns in the cages, where they were able to move freely. Knee joint swelling severity and limping were classified as no change (0), mild (1), moderate (2), and severe (3) [17]. All assessments were conducted by the same trained inspector who was blinded to their treatments throughout the study period. 

Pain-related behaviors were assessed using an incapacitance test via a hind paw limb weight-bearing apparatus (Linton Incapacitance Tester, Linton Instruments, Palgrave Diss, UK), the maximum running speed on a treadmill, and locomotive activity. These assessments occurred on the same days as OA progression was assessed. These tests have been utilized as indices of joint discomfort and may be useful for the discovery of novel pharmacological agents for treating human OA [17]. Rats were acclimatized for 30 min before the incapacitance test. The hind paw weight distribution between the right (osteoarthritis) and left (control) limbs were assessed using an incapacitance tester. The assessments were performed five times for each rat, and the average of their values was calculated. The percent weight distribution of the right hind paw was calculated by dividing the right paw weight by the sum of the left and right paw weights [18]. 

The maximum running speed of the rats was used as a parameter to assess the severity of their OA, as rats with OA cannot run as fast as rats without OA. The rats walked on a treadmill at 40 cm/s for 1 min, which then increased to 50 cm/sec for 1 min. Subsequently, the speed of the treadmill increased by 5 cm/s every min until the rats could not continue to run and instead slid to the back of the treadmill. The maximum running speed for each rat was determined by them running for 20 s at that speed. Each rat was subjected to the treadmill test for under 5 min. 

The locomotive activity was measured using a Linton AM1053 Activity Monitor system to mount a three-dimensional array of infrared beams around clear Perspex cages with AmLogger software (Linton Instruments). The total locomotive activity was calculated through the sum of the rearing, mobility, and activities measured by the infrared beams. The activity was measured for 30 min during the dark phase of the light/dark cycle when the rats were most active after they were adapted to the clear Perspex cage for 30 min. 

### 2.10. Glucose Homeostasis and Sample Collection at the end of the Experiment 

An oral glucose tolerance test (OGTT) was performed via the overnight fasting of rats for 16 h, followed by administering 2 g glucose/kg bw orally at week eleven. Post-glucose loading, tail blood was collected to measure serum glucose levels using a Glucose Analyzer II (Beckman, Palo Alto, CA, USA) at 10 min intervals from 0 to 90 min and at 120 min. At 0, 20, 40, 90, and 120 min serum insulin concentrations were determined using an Ultrasensitive ELISA kit (Linco Research, Billerica, MA, USA). The average of the areas under the curves (AUC) of the serum glucose and insulin concentrations was calculated using the trapezoidal rule. 

The rats were anesthetized with a ketamine/xylazine mixture, and the peri-uterine and retroperitoneal fat masses and uterine were weighed after excision. The uterus index (uterus weight divided by body weight) was calculated. Insulin resistance was determined using the homeostasis model assessment estimate of insulin resistance (HOMA-IR). HOMA-IR was calculated by fasting insulin (µIU/mL) × fasting glucose (mM)/22.5. The serum was prepared from blood collected from the inferior vena cava by centrifuging at 3000 rpm for 20 min. Serum and tissues were then stored at −70 °C for future use. Serum IL-6 and TNF-α concentrations were determined using a commercially available ELISA (Rat IL-6 and Rat TNF-α Quantikine (R&D Systems, Minneapolis, MN, USA). Serum 17β-estradiol levels and serum alkaline phosphatase (ALP) activities were measured by ELISA kits (Enzo Life Sciences, Farmingdale, NY, USA) and colorimetry kit (Asan Pharmaceutical, Seoul, Korea).

### 2.11. Isolation of Total RNA from Articular Cartilage and Real-Time PCR 

Articular cartilage samples from five rats of each group were collected at the end of the experiment. Each cartilage was individually powdered with a cold steel mortar and pestle and then mixed with a monophasic solution of phenol and guanidine isothiocyanate (TRIzol reagent, Life Technologies, Rockville, MD, USA) for total RNA extraction. The RNA concentration was determined using a Lambda 850 spectrophotometer (Perkin Elmer, Waltham, MA, USA) and cDNA was synthesized from 1 μg total RNA extracted from individual rats using a Superscript III reverse transcriptase kit (Life Technology, Carlsbad, CA, USA). Five different cDNA were formed from each group, and each cDNA was used for real-time PCR. Equal amounts of cDNA and primers for the gene of interest were mixed with SYBR Green mix (Bio-Rad, Richmond, CA, USA) in duplicate and amplified using a real-time PCR instrument (Bio-Rad). Thermal cycling conditions were 55 °C for 2 min and 95 °C for 10 min, followed by 40 cycles of 94 °C for 20 s, 65 °C for 30 s, and 72 °C for 20 seconds. Primers were used to detect the genes related to the inflammation and degradation of articular cartilage, such as TNF-α, IL-1β, IL-6, matrix metalloproteinase (MMP)-3, and MMP-13 genes. The primers were described previously [18]. The cycle of threshold (CT) for each sample was determined via real-time PCR. The gene expression levels in the cartilage of each rat were quantitated using the CT method (ΔΔCT method). ΔCT was calculated via subtracting CT (target gene) to CT (endogenous reference gene, β-actin). The relative fold-change in gene expression was calculated by the equation ΔΔCt = ΔCt_treatment_ − ΔCt_control_. The results are presented as 2^−ΔΔCT^ [16].

### 2.12. Histopathological Analysis of Rat Knees 

After scarifying rats, the right knee was histologically examined for chronic morphological changes, including narrowing knee articular bones, the loss of joint regions, cartilage erosion, and osteophyte formation. For the histological analysis, knee joints were excised and fixed in phosphate-buffered formalin. The joints were subsequently decalcified in 10% nitric acid for 72 h and embedded in paraffin. Five-micrometer sections were stained with hematoxylin and eosin (H-E) and safranin-O fast green to evaluate morphological changes. The histopathological changes in each rat were quantitatively expressed according to the depth and extent of the damage by the following scoring system [17,18]: The depth was scored on a scale of 0–5 where 0 = normal; 1 = minimal, affecting the superficial zone only; 2 = mild invasion into the upper middle zone only; 3 = moderate invasion well into the middle zone; 4 = marked invasion into the deep zone but not to the tidemark; and 5 = severe full-thickness degradation to the tidemark. The extent of the tibial plateau involvement and proteoglycan loss were scored as 1 (minimal), 2 (mild), 3 (moderate), and 4 (severe). 

### 2.13. Statistical Analysis 

Statistical analysis was performed using SAS software version 7 (SAS Institute, Cary, NC, USA), and all results are expressed as a mean±SD. The variables measured at different time points were analyzed via two-way repeated measures ANOVA with the time and group as independent variables and an interaction term between the time and group. A one-way ANOVA was used to determine the metabolic effects of the OVX (OVX rats with a saline injection), OA (OVA rats with an MIA injection), OVX-OA-RAFR-L, OVX-OA-RAFR-H, and Sham (Sham rats with a saline injection; normal-control). Significant differences in the effects between groups were identified through Tukey’s test at *p* < 0.05. 

## 3. Results

### 3.1. Index Compounds of RAFR 

RAFR contained lysine as an index compound for rice and alliin, s-allyl-L-cysteine, γ-glutamyl-(s)-allyl-cysteine, ferulic acid, and quercetin as the index compounds for AFR (Table 1, Figure 1).

### 3.2. Menopausal Symptoms

Uterine weights and serum 17β-estradiol levels at the end of the experimental periods were significantly lower in the OVX rats than in the Sham rats. However, these levels were not significantly different among all OVX groups (Table 2). Tail skin temperatures, an index of a hot flush, increased at eight and eleven weeks in the OVX rats compared to that in the Sham rats. Tail skin temperatures were not significantly different between the OVX and OVX-OA groups. RAFR did not modulate tail skin temperatures in OVX-OA rats (Table 2). 

### 3.3. Energy Metabolism

OVX rats experienced greater body weight gains during the first seven weeks than the Sham rats. Before MIA injection, food intake was not significantly different OVX and OVX-OA groups (Table 3). However, a decrease in body weight gains in the OVX-OA group compared to the OVX group. After MIA injection food intake was lower in the OVX-OA group than in the OVX group. Thus, the decrease in weight gain in the OVX-OA rats was associated with decreased food intake in the OVX-OA group (Table 3). The OVX-OA-RAFR-L and OVX-OA-RAFL-H did not experience a decrease in body weight gains, but the increase was lesser than that of the OVX group (Table 3). The food intake in the OVX-OA-RAFR-L and OVX-OA-RAFR-H did not differ from the OVX and Sham groups (Table 3). Therefore, MIA injection into the knee joints decreased food intake, possibly due to pain from developing OA. RAFR intake prevented a decrease in food intake, indicating that RAFR intake may decrease this pain. Visceral fat contents measured via the peri-uterine and retroperitoneum fat mass were much higher in the OVX than in the Sham (Table 3). Although body weight gains were lower in the OVX-OA group than the OVX group, visceral fat masses were not significantly different between the OVX-OA and OVX groups (Table 3). The OVX-OA-RAFR decreased the fat masses in a dose-dependent manner more than that in the OVX-OA.

### 3.4. Glucose Metabolism 

Serum glucose levels at the fasting state were higher in the OVX and OVX-OA than in the Sham. The OVX-OA-RAFR lowered the concentrations more than the OVX-OA in a dose-dependent manner (Table 4). The OVX-OA-RAFR-H decreased the concentrations by the same degree as the Sham. Serum insulin concentrations were higher in the OVX and OVX-OA compared to those in the Sham. The OVX-OA-RAFR-H lowered serum insulin concentrations than the Sham. HOMA-IR, an index of insulin resistance, increased in the OVX and OVX-OA compared to that in the Sham. The OVX-OA-RAFR decreased this index dose-dependently (Table 4). After the oral glucose challenge, serum glucose concentrations were elevated for 30–40 min then slowly reduced in all rats (Figure 2A). OVX and OVX-OA rats had higher peaks at 30 min than the Sham rats, whereas the RAFR treatments lowered the peak more than the OVX and OVX-OA. The OVX-OA-RAFR-H decreased serum glucose concentrations compared to the Sham at 30–90 min (Figure 2A). The AUC of glucose in the first and second parts of OGTT was higher in the OVX and OVX-OA compared to that in the Sham. The AUC in the first and second parts was lowered in the OVX-OA-RAFR-L and OVX-OA-RAFR-H compared to that in the OVX and OVX-OA, and it was similar to that in the Sham (Figure 2B). 

### 3.5. Body Composition in the Hip and Right Leg

After eleven weeks of OVX, BMD of the lumbar spine and non-OA leg was lower in the OVX and OVX-OA compared to that in the Sham. The BMD was also significantly increased in the OVX-OA-RAFR-H group compared to that in the OVX-OA group (Figure 3A). However, BMD in the OA leg was not significantly different in all OA groups (Figure 3A).

These results indicate that while OVX reduced the BMD, the OVX-OA did not further decrease it within three weeks. Serum ALP activity was higher in the OVX and OA than in the Sham, and the OVX-OA-RAFR-H lowered this activity to the same extent as the Sham (Table 4). Therefore, the OVX activated bone catabolism to a greater degree than the normal control, but the OVX-OA did not stimulate bone catabolism. The OVX-OA-RAFR inhibited this catabolism. 

At the end of the experiment, LBM was lower in all OVX rats compared to that in the Sham rats, and the OVX-OA did not modulate LBM in OVX rats (Figure 3B). The OVX-OA-RAFR suppressed the decrease of LBM in the hips and non-OA legs dose-dependently, but the OVX-OA-RAFR did not exhibit the dose-dependent inhibition of an LBM decrease in the OA legs. 

After eleven weeks, the fat mass in the abdomen and legs was higher in the OVX than in the Sham, whereas the OVX-OA lowered the fat mass in the abdomen but increased it in the OA legs than the OVX (Figure 3C). RAFR decreased in the abdomen and both non-OA and OA legs compared to the OVX-OA. The fat mass in the abdomen and legs decreased OVX-OA-RAFR-H to the same degree as the Sham (Figure 3C). 

### 3.6. Global OA Symptoms and Pain-Related Behaviors 

The rats who received an MIA injection developed global OA symptoms such as swelling and limping, and they were assessed weekly. The swelling scores were greater in the OVX-OA group than in all other groups, and they gradually decreased as time passed (Figure 4A). RAFR lowered the swelling scores more than the OA in a dose-dependent manner. However, the OVX-OA-RAFR-H did not improve the scores as much as the OVX. The limping scores show a similar pattern to the swelling scores. Increased limping scores due to OA were dose-dependently reduced by RAFR (Figure 4B).

OA is accompanied by pain, the severity of which was mainly determined by the maximum velocity of running on a treadmill and asymmetric weight distribution. The weight distributions in the OA-induced knees were much lower than in the non-OA-induced knees due to pain induced by OA during the first week after MIA injection (Figure 4C). The OVX-OA-RAFR-H ameliorated pain by OA after two weeks of MIA injection as time passed (Figure 4C). The maximum running velocities were higher in the OVX than in the Sham, indicating that menopause decreased the maximum running velocity of the rats without an OA injection (Fig. 4D). The OVX-OA-RAFR-H increased the maximum running velocity compared to the OVX-OA but it decreased compared to the OVX until day 14. However, at day 21, the maximum running velocity in the OVX-OA-RAFR-L was elevated to the same degree as the OVX (Figure 4D). 

### 3.7. The mRNA Expression of Cytokines in the Articular Cartilage of the Right Knee 

MIA stimulates MMPs known as collagenases that degrade extracellular matrix containing collagen and release proinflammatory cytokines in the knee joint. The expression of MMP-1 and MMP-13 in the articular cartilage increased in the OVX rats compared to that in the Sham and OVX-OA rats (Figure 5A). The OVX-OA-RAFR-L and OVX-OA-RAFR-H decreased the fold increase of MMP-1 and MMP-13 mRNA expression dose-dependently. The OVX-OA-RAFR-H decreased MMP-13 as much as the normal-control (Figure 5A). 

Like MMP-1 and MMP-13, the OVX rats showed increased expression of pro-inflammatory cytokines such as TNF-α, IL-1β, and IL-6 in articular cartilage in comparison to the Sham rats (Figure 5B). The fold increase of their mRNA expression was much higher in the OVX-OA than in the OVX. The OVX-OA-RAFR-L and OVX-OA-RAFR-H decreased the mRNA expression of proinflammatory cytokines dose-dependently. However, the decrease was not to the same extent as the Sham.

Furthermore, serum IL-6 and TNF-α concentrations were elevated in the OVX rats compared to that in the Sham rats. The OVX-OA rats showed increased IL-6 and TNF-α concentrations in comparison to the OVX rats (Figure 5C). The OVX-OA-RAFR lowered serum IL-6 and TNF-α compared to the OVX-OA in a dose-dependent manner (Figure 5C).

### 3.8. Histopathological Analysis 

In histological evaluations using H-E staining, the Sham and OVX rats exhibited normal articular cartilage structures with smooth articular surfaces, normal chondrocytes with columnar orientation, and intact tide marks and subchondral bone. After MIA injection into the knees, the OVX-OA rats induced the degeneration of columnar orientation, degeneration of the tide mark, and the penetration of subchondral bones (Figure 6A). The OVX-OA-RAFR delayed the exacerbation of the articular structure degeneration, reduced the penetration of the subchondral bones, and reduced the degeneration of the tide marks in comparison to the OVX-OA (Figure 6A). Safranin-O fast green staining revealed that the OVX-OA-RAFR dose-dependently decreased the loss of proteoglycan in the knee joints compared to the OVX-OA (Figure 6B). However, the OVX-OA-RAFR group did not improve the morphology of the articular cartilage as much as the OVX group (Figure 6B). These results indicate that RAFR treatment partially prevented the breakdown of the articular surface in OVX rats with MIA injection into the right knee.

## 4. Discussion

Estrogen deficiency during menopause exacerbates bone, energy, glucose, and lipid metabolism by increasing insulin resistance. These effects are accompanied by elevated oxidative stress and inflammation. Women experiencing menopause have a high susceptibility to OA. Moreover, hyperglycemia exacerbates the symptoms of OA. In the present study, we examined whether RAFR ameliorated bone and glucose metabolism involved in OA progression as well as menopausal symptoms in OVX rats. RAFR did not decrease tail skin temperature, uterine weight, or serum 17β-estradiol concentrations compared to OVX-control rats. However, RAFR ameliorated BMD and LBM, but not to the same extent as the Sham.

Furthermore, OVX-OA-RAFR improved glucose and insulin tolerance to a similar degree as the Sham. This improvement is associated with the exacerbation of the activation of MMPs to reduce collagen and stimulate proinflammatory cytokines due to an estrogen deficiency. This study shows that OVX-OA-RAFR reduced the disturbance of glucose and bone metabolism but did not significantly decrease tail skin temperatures in OVX-OA rats. MIA is known to disrupt glycolysis in the knee joints resulting in the eventual death of chondrocytes by inhibiting glyceraldehyde-3-phosphate dehydrogenase activity with increased inflammation [20]. RAFR dose-dependently ameliorated OA symptoms induced by MIA in the knee joint by decreasing the degradation of articular cartilage and the mRNA expression of proinflammatory cytokines. Therefore, it may be beneficial for elderly women who are at risk of OA to consume RAFR daily. 

AFR, a part of RAFR, contains Allium sulfides such as alliin and allicin [21]. AFR inhibits cholesterol synthesis, consists of arterial smooth muscle cells, and exhibits platelet aggregation as well as anti-oxidant, anti-inflammatory, and vasodilatory activities [21]. Previous studies have reported that AFR reduces body fat and liver fat in non-alcoholic fatty livers [22,23]. However, no previous studies have reported the impact of AFR on OA and menopausal symptoms. Allium sulfide compounds have a great potential to contribute towards treatments for OA. Allium sulfide has been demonstrated to repress the expression of matrix-degrading proteases in chondrocyte-like cells, meaning that it may have the potential to reduce OA incidence [24]. The present study found that OVX-OA-RAFR decreases the mRNA expression of MMP-1 and MMP-13, which are involved with the matrix-degrading proteases, and TNF-α and IL-13 in the articular matrix compared to the OVX-OA. These effects may be associated with allium sulfide in RAFR. 

A decline in ovarian function is associated with a spontaneous increase in proinflammatory cytokines along with oxidative stress. This outcome is associated with the exacerbation of osteoarthritis symptoms in menopausal women. In the estrogen-deficient state, nuclear factor kappa B is activated, which increases the production of IL-1β, IL-6, and TNF-α. In OVX rats, their concentrations and mRNA expression are elevated, which suggests the exacerbation of OA due to estrogen deficiency [25]. The present study showed consistent results; OVX rats decreased BMD and LBM with increasing serum TNF-α concentrations. The serum TNF-α concentrations were elevated in OVX-OA rats compared to OVX rats.

Moreover, the mRNA expressions of IL-1β, IL-6, and TNF-α in the articular matrix in the knee were elevated in OVX-OA rats compared to OVX rats. No studies have been conducted on the effects of AFR or RAFR on bone metabolism. However, previous studies have found that the garlic extract maintains bone health in OVX rats. The reduction of estrogen production increases bone turnover by increasing the production of proinflammatory cytokines and oxidative stress to induce osteoporosis [25]. Garlic extracts protect against BMD loss [25]. The effect of RAFR on BMD may be similar to that of garlic, as RAFR contains allium sulfide compounds which may be effective components. This present study showed that RAFR protected against a decrease of BMD and LBM by decreasing the mRNA expression of IL-1β, IL-6, and TNF-α in the articular matrix of the knee joint. 

OA changes the morphology of the knee joints, and these changes can be scored to assess OA severity. The induction of OA increases chondrocyte hypertrophy and decreases the articular cartilage matrix with the loss of proteoglycans and surface fibrillation [26]. In addition, the early stages of OA increase bone remodeling and bone loss, and the late-stage slows down remodeling and subchondral densification. Therefore, OA bone atrophy arises from an increase in bone resorption or a decrease in bone formation during OA [27]. In this present study, MIA reduces the articular cartilage matrix via the upregulation of matrix-degrading enzymes such as MMP-1 and MMP-13 and bone loss. Bone loss was indicated via a higher serum ALP concentration and lower BMD in the legs of the OA-control group compared to that in the OVX-control group. OVX-OA-RAFR reduced proteoglycan loss in the articular cartilage matrix as well as bone atrophy and BMD loss compared to OVX-OA. RAFR partly protected against the loss of the articular cartilage matrix and bone remodeling in MIA-induced OA rats. 

## 5. Conclusions

OVX-OA-RAFR improved both glucose and bone metabolism in OVX-OA rats to a degree almost similar to the sham rats. OVX-OA-RAFR protected against the decrease of BMD and LBM in the legs and increased the mass of fat in the abdomen. OVX-OA-RAFR dose-dependently protected against pain-related behaviors in OVX-OA. The OVX-OA deteriorated the articular cartilage by reducing collagen and bone loss in the knee joints and OVX-OA-RAFR prevented their deterioration in OA-induced rats. The improvements by RAFR are associated with a decrease in the mRNA expression of *MMP3* and *MMP13* and proinflammatory cytokines such as *TNF-α*, *IL-1β*, and *IL-6* in the articular cartilage of OVX-OA rats. These results suggest that daily intake of RAFR is effective for treating OA symptoms in OA-induced estrogen-deficient rats and may have a potential therapeutic effect for treating OA in menopausal women.

## Figures and Tables

**Figure 1 nutrients-11-01503-f001:**
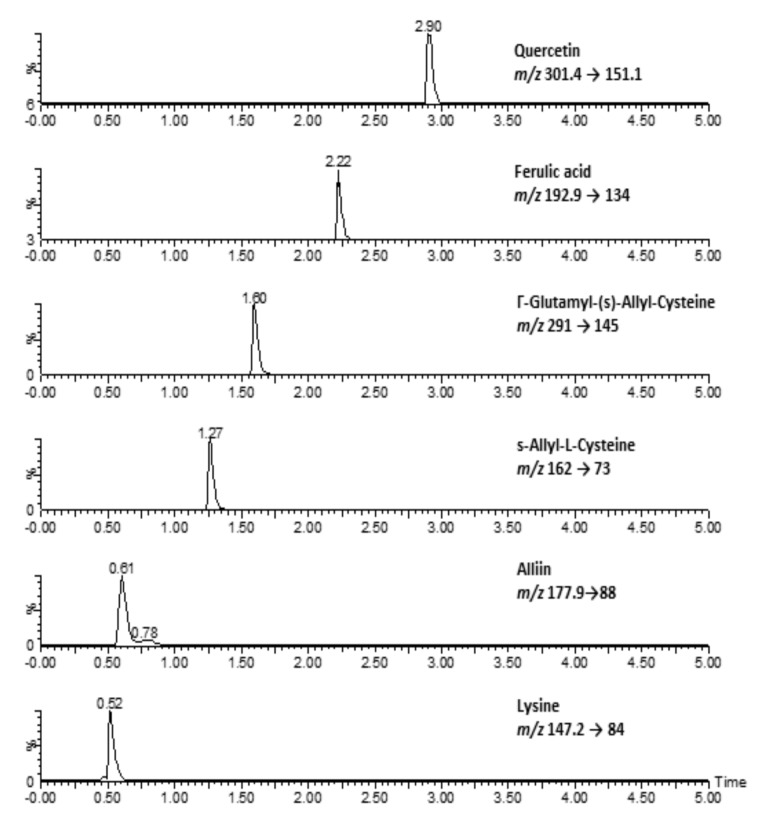
The UPLC-MS/MS chromatograms of rice porridge containing *Allium fistulosum* root water extract.

**Figure 2 nutrients-11-01503-f002:**
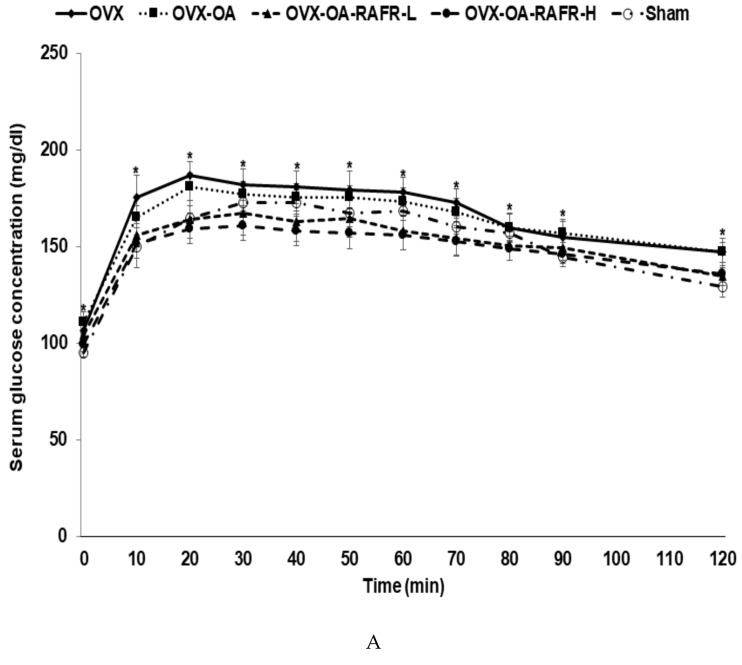
The area under the curve (AUC) of serum glucose concentrations during the oral glucose tolerance test. Ovariectomized (OVX) rats were divided into four groups: (1) MIA injection into the knee joint and fed a high-fat diet containing 0.23% rice + 0.018% AFR diet (RAFR-L). (2) MIA injection into the knee joint and fed a high-fat diet containing 0.69% rice + 0.053% AFR diet (RAFR-H). (3) MIA injection into the knee joint and fed a high-fat diet containing 0.053% cellulose (OA-control). (4) saline injection into the knee joint and fed a high-fat diet containing 0.053% cellulose (non-OA control). Sham rats had the same diet as a control and received a saline injection (normal-control). At the beginning of the eighth week, an articular injection of monoiodoacetate (MIA) into the right knee was performed in all OVX groups except the non-OA control and normal-control groups. Two weeks after the MIA injection, oral glucose tolerance tests were performed with 2 g glucose per kg body weight after 16 h of fasting. Serum glucose (**A**) was measured, and the area under the glucose curve (**B**) was calculated. The dots or bars and error bars represent the mean+SD (*n* = 10). ^a,b^ Values of the bars with different superscripts were significantly different among groups, as per the Tukey test at *p* < 0.05.

**Figure 3 nutrients-11-01503-f003:**
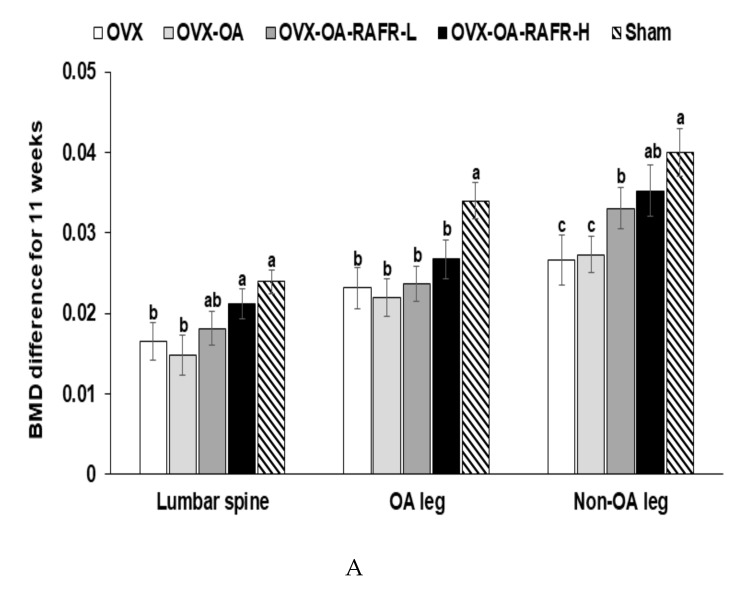
Bone mineral density (BMD) and the lean mass of the femur and knee with the intra-articular injection of monoiodoacetate (MIA) at days 0 and 21 after MIA-injection Ovariectomized (OVX) rats were divided into four groups: (1) MIA injection into the knee joint and fed a high-fat diet containing 0.23% rice + 0.018% AFR diet (RAFR-L). (2) MIA injection into the knee joint and fed a high-fat diet containing 0.69% rice + 0.053% AFR diet (RAFR-H). (3) MIA injection into the knee joint and fed a high-fat diet containing 0.053% cellulose (OA-control). (4) saline injection into the knee joint and fed a high-fat diet containing 0.053% cellulose (non-OA control). Sham rats had the same diet as the control and received a saline injection (normal-control). At the beginning of the eighth week, an articular injection of monoiodoacetate into the right knee was performed in all OVX groups, except the normal-control group, and the assigned diets were provided for an additional three weeks. The BMD (**A**) and lean body mass (**B**) of the hips and right legs, as well as the mass of the fat in the abdomens and right legs (**C**), were measured via DEXA. Each bar and error bar represents the mean *±* SD (*n* = 10). ^a,b,c,d^ Values of the bars with different superscripts were significantly different among groups as per the Tukey test at *p* < 0.05.

**Figure 4 nutrients-11-01503-f004:**
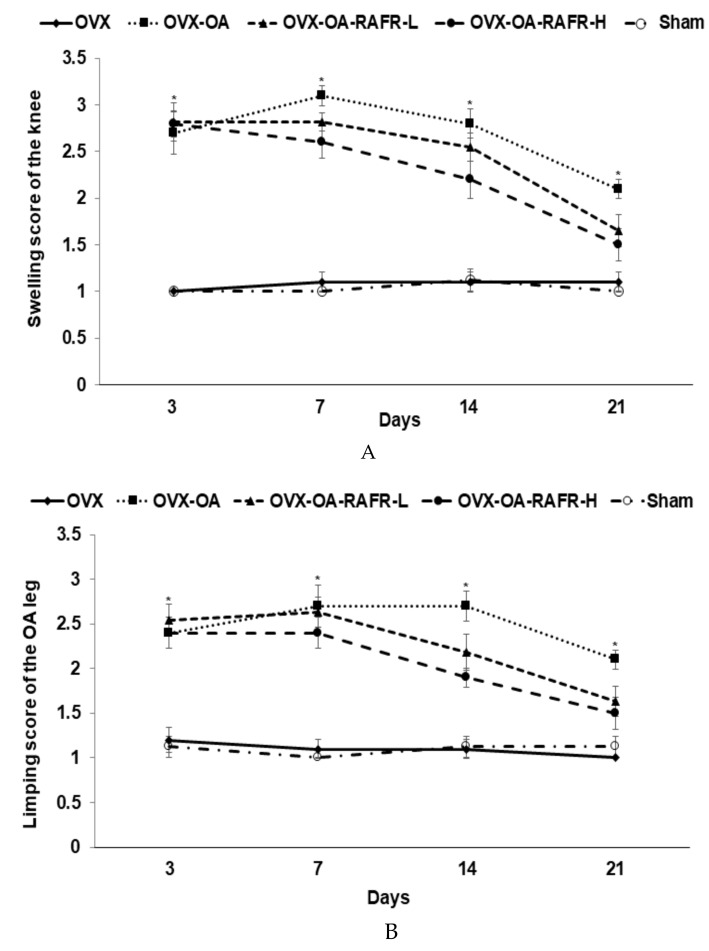
Gross observations of osteoarthritis symptoms and pain-related behaviors at 3, 7, 14, and 21 days after monoiodoacetate (MIA)-injection Ovariectomized (OVX) rats were divided into four groups: (1) MIA injection into the knee joint and fed a high-fat diet containing 0.23% rice + 0.018% AFR diet (RAFR-L). (2) MIA injection into the knee joint and fed a high-fat diet containing 0.69% rice + 0.053% AFR diet (RAFR-H). (3) MIA injection into the knee joint and fed a high-fat diet containing 0.053% cellulose (OA-control). (4) Saline injection into the knee joint and fed a high-fat diet containing 0.053% cellulose (non-OA control). Sham rats had the same diet as the control and received a saline injection (normal-control). At the beginning of the eighth week, an articular injection of monoiodoacetate (MIA) into the right knee was performed in all OVX groups except the normal-control group. The assigned diets were provided for an additional three weeks. During the gross observations of osteoarthritis symptoms, the swelling (**A**) and limping (**B**) scores in the right knee were measured. Differences in the weight distribution of the right hind paws (**C**) were measured via an incapacitance tester and the maximum running velocity on a treadmill (**D**) as indicators of knee pain. Each data point and error bar represent the mean *±* SD (*n* = 10). * Significant treatment effect by repeated measures of a two-way ANOVA test at *p* < 0.05. ^‡^ Significant time effect by repeated measures of a two-way ANOVA test at *p* < 0.05. ^a,b,c,d^ Values of the bars with different superscripts were significantly different among groups as per the Tukey test at *p* < 0.05.

**Figure 5 nutrients-11-01503-f005:**
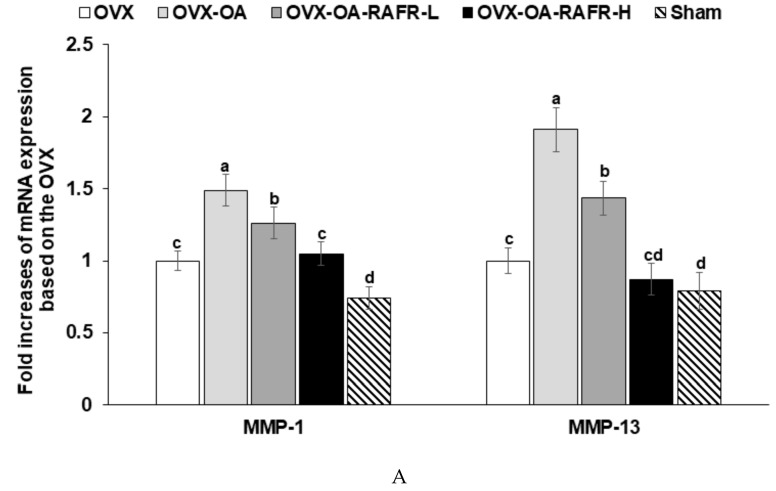
The mRNA expression of matrix metalloproteinases and pro-inflammatory cytokines in the articular cartilage after 21 days of intra-articular injection of monoiodoacetate (MIA) Ovariectomized (OVX) rats were divided into four groups: (1) MIA injection into the knee joint and fed a high-fat diet containing 0.23% rice + 0.018% AFR diet (RAFR-L). (2) MIA injection into the knee joint and fed a high-fat diet containing 0.69% rice + 0.053% AFR diet (RAFR-H), (3) MIA injection into the knee joint and fed a high-fat diet containing 0.053% cellulose (OA-control). (4) saline injection into the knee joint and fed a high-fat diet containing 0.053% cellulose (non-OA control). Sham rats had the same diet as the control and received a saline injection (normal-control). At the beginning of the eighth week, an articular injection of monoiodoacetate into the right knee was performed in all OVX groups except the normal-control group, and the assigned diets were provided for an additional three weeks. The mRNA expression of MMP-13 and MMP-13 involved in collagen degradation (**A**) and the cytokines (TNF-α, IL-1β, and IL-6) that result in inflammation (**B**) were measured via real-time PCR. Serum proinflammatory cytokines (TNF-α and IL-6) were measured by ELISA (**C**). Each bar and error bar represents the mean *±* SD (*n* = 6). ^a,b,c,d,e^ Different letters indicate significant differences in the treatment groups of OVX rats at each time point identified by Tukey’s test at *p* < 0.05.

**Figure 6 nutrients-11-01503-f006:**
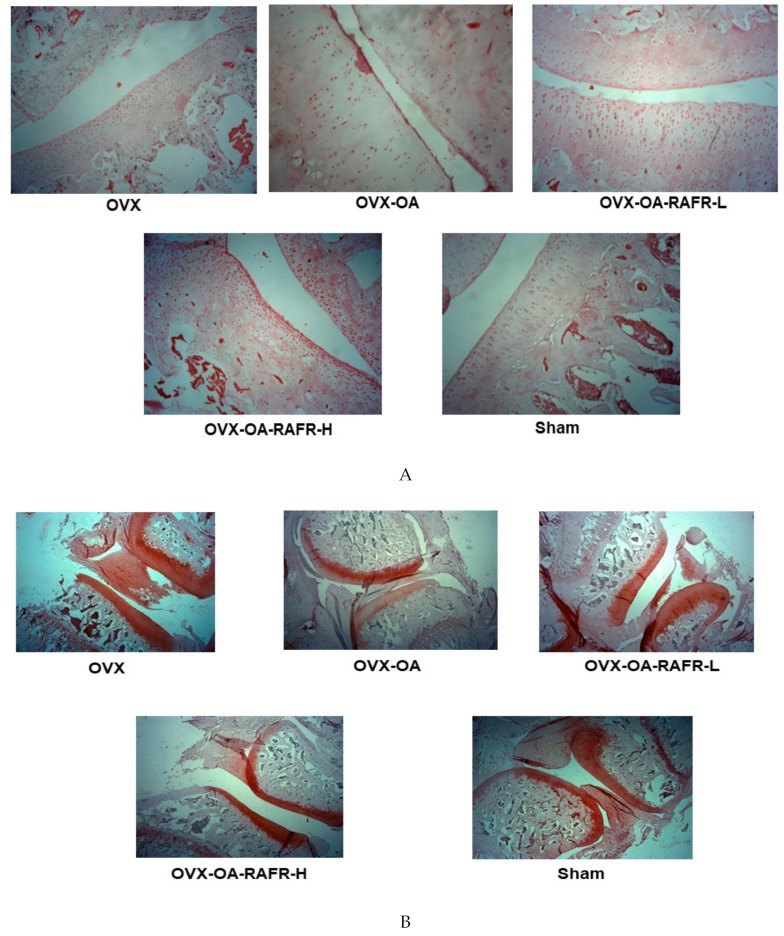
The histopathological features of osteoarthritic lesions in the knee joints of rats after 21 days of intra-articular injection of monoiodoacetate (MIA) Ovariectomized (OVX) rats were divided into four groups: (1) MIA injection into the knee joint and fed a high-fat diet containing 0.23% rice + 0.018% AFR diet (RAFR-L). (2) MIA injection into the knee joint and fed a high-fat diet containing 0.69% rice + 0.053% AFR diet (RAFR-H). (3) MIA injection into the knee joint and fed a high-fat diet added 0.053% cellulose (OA-control). (4) saline injection into the knee joint and fed a high-fat diet containing 0.053% cellulose (non-OA control). Sham rats had the same diet as the control and received a saline injection (normal-control). At the beginning of the eighth week, an articular injection of monoiodoacetate into the right knee was performed in all OVX groups except the normal-control group, and the assigned diets were provided for an additional three weeks. The depth and extent of cartilage damage and the quantification of the damage were determined in hematoxylin-eosin stained (**A**), paraffin-embedded knee joint sections from MIA-injected rats (magnifying power ×10). The depth and extent of cartilage damage and quantification of the cartilage damage in the MIA-injected knees were evaluated in Safranin O–fast green–stained (**B**) knee joint sections from the osteoarthritic rats (magnifying power ×2.5). Scores of the knee joint damage (**C**) were calculated from the stained sections. Each bar and error bar represents the mean *±* SD (*n* = 5). ^a,b,c,d^ Values of the bars with different superscripts were significantly different among groups as per the Tukey test at *p* < 0.05.

**Table 1 nutrients-11-01503-t001:** Index compounds of rice porridge containing *Allium fistulosum* root extracts.

Index Compounds	Amount (ug/g Dry Weight)
Lysine	109.1 ± 5.51
Alliin	47.5 ± 0.0
s-Allyl-L-cysteine	0.59 ± 0.09
γ-Glutamyl-(s)-allyl-cysteine	4.22 ± 0.29
Ferulic acid	2.90 ± 0.0
Quercetin	3.69 ± 0.0

**Table 2 nutrients-11-01503-t002:** Parameters related to the menopausal symptoms.

Parameters	OVX (*n* = 10)	OVX-OA (*n* = 10)	OVX-OA-RARF-L (*n* = 10)	OVX-OA-RARF-H, (*n* = 10)	Sham (*n* = 10)
Uterine weight (g)	0.18 ± 0.06 ^b^	0.17 ± 0.06 ^b^	0.18 ± 0.05 ^b^	0.18 ± 0.06 ^b^	0.51 ± 0.08 ^a^
Serum 17β-estradiol levels (pg/mL)	1.7 ± 0.7 ^b^	1.6 ± 0.6 ^b^	1.7 ± 0.6 ^b^	1.6 ± 0.7 ^b^	7.2 ± 1.0 ^a^
Skin temperature (°C) at 8 weeks	29.4 ± 0.05 ^a^	29.3 ± 0.05 ^a^	28.9 ± 0.03 ^ab^	29.0 ± 0.03 ^ab^	28.6 ± 0.08 ^b^
Skin temperature (°C) at 11 weeks	29.5 ± 0.09 ^a^	29.4 ± 0.15 ^a^	28.9 ± 0.09 ^ab^	29.0 ± 0.09 ^ab^	28.4 ± 0.08 ^b^

Ovariectomized rats were divided into four groups: (1) monoiodoacetate (MIA) injection into the knee joint and fed a high fat diet containing 0.23% rice+0.018% AFR diet (OVX-OA-RAFR-L), (2) MIA injection into the knee joint and fed a high fat diet containing 0.69% rice+0.053% AFR diet (OVX-OA-RAFR-H), (3) MIA injection into the knee joint and fed a high fat diet added 0.053% cellulose (OVX-OA), or (4) saline injection into the knee joint and fed a high fat diet containing 0.053% cellulose (OVX). Sham rats had the same diet as control and saline injection (Sham; normal-control). At the beginning of the 8th week, an articular injection of monoiodoacetate into the right knee was performed in all OVX groups except the normal-control group and the assigned diets were provided for an additional 3 weeks. Values are mean ± SD. ^a,b^ Values on the same row with different superscripts were significantly different among groups by Tukey test at *p* < 0.05.

**Table 3 nutrients-11-01503-t003:** Metabolic parameters at the end of experimental periods.

Metabolic Parameters	OVX (*n* = 10)	OVX-OA (*n* = 10)	OVX-OA-RARF-L (*n* = 10)	OVX-OA-RARF-H, (*n* = 10)	Sham (*n* = 10)
Body weight gain for the first 8 weeks prior to MIA injection	118 ± 12 ^a^	117 ± 12 ^a^	124 ± 9 ^a^	118 ± 20 ^a^	74.6 ± 11 ^b^
Body weight gain after MIA injection	8.3 ± 1.1 ^a^	−4.3 ± 3.4 ^d^	5.9 ± 1.2 ^b^	5.1 ± 1.3 ^b^	2.9 ± 0.5 ^c^
Body weight gain for 11 weeks	127 ± 14 ^a^	113 ± 13 ^b^	130 ± 1.2 ^a^	123 ± 1.5 ^ab^	77.5 ± 10 ^c^
Food intake at 8th week (g)	11.5 ± 1.1	11.3 ± 1.2	10.9 ± 1.7	11.8 ± 1.6	11.6 ± 1.1
Food intake at 11th week (g)	11.7 ± 1.6 ^a^	9.2 ± 0.9 ^b^	10.5 ± 1.4 ^ab^	10.7 ± 1.1 ^ab^	11.0 ± 1.3 ^a^
Average intake of RARF (mg/day)	0	0	57.4 ± 6.5	55.6 ± 6.1	0
Peri-uterine fat (g)	12.5 ± 1.5 ^ab^	13.4 ± 1.4 ^a^	11.7 ± 1.2 ^b^	9.5 ± 1.1 ^c^	6.8 ± 0.9 ^d^
Retroperitoneum fat (g)	8.5 ± 1.1 ^a^	8.3 ± 0.8 ^a^	7.0 ± 0.7 ^b^	6.2 ± 0.7 ^c^	5.1 ± 0.8 ^d^
Visceral fat/body weight (%)	6.1 ± 0.7 ^a^	6.4 ± 0.6 ^a^	5.0 ± 0.6 ^b^	4.4 ± 0.6 ^c^	3.9 ± 0.6 ^c^

Ovariectomized rats were divided into four groups: (1) monoiodoacetate (MIA) injection into the knee joint and fed a high fat diet containing 0.23% rice+0.018% AFR diet (OVX-OA-RAFR-L), (2) MIA injection into the knee joint and fed a high fat diet containing 0.69% rice+0.053% AFR diet (OVX-OA-RAFR-H), (3) MIA injection into the knee joint and fed a high fat diet added 0.053% cellulose (OVX-OA), or (4) saline injection into the knee joint and fed a high fat diet containing 0.053% cellulose (OVX). Sham rats had the same diet as control and saline injection (Sham; normal-control). At the beginning of the 8^th^ week, an articular injection of monoiodoacetate into the right knee was performed in all OVX groups except the normal-control group and the assigned diets were provided for an additional 3 weeks. Values are mean ± SD. ^a,b,c,d^ Values on the same row with different superscripts were significantly different among groups by Tukey test at *p* < 0.05.

**Table 4 nutrients-11-01503-t004:** Serum glucose, insulin and triglyceride concentrations.

	OVX (*n* = 10)	OVX-OA (*n* = 10)	OVX-OA-RARF-L (*n* = 10)	OVX-OA-RARF-H, (*n* = 10)	Sham (*n* = 10)
Serum alkaline phosphatase activity	71.9 ±10.3 ^a^	74.3 ± 10.4 ^a^	60.5 ± 9.7 ^b^	48.0 ± 5.7 ^c^	51.5 ± 5.6 ^c^
Serum glucose (mg/dL)	106 ± 8 ^ab^	110 ± 7 ^a^	104 ± 7 ^ab^	99.4 ± 7 ^b^	95.2 ± 6 ^b^
Serum insulin (ng/mL)	1.87 ± 0.16 ^a^	1.85 ± 0.14 ^b^	1.37 ± 0.15 ^b^	1.19 ± 0.11 ^c^	1.33 ± 0.15 ^a^
HOMA-IR	8.9 ± 0.9 ^a^	9.1 ± 0.8 ^a^	6.3 ± 0.7 ^b^	5.3 ± 0.6 ^c^	5.6 ± 0.7 ^c^
Serum triglyceride (mg/dL)	105.9 ± 8.4 ^a^	110 ± 8.8 ^a^	98.6 ± 7.5 ^b^	90.4 ± 8.3 ^c^	86.8 ± 6.6 ^c^

Ovariectomized rats were divided into four groups: (1) monoiodoacetate (MIA) injection into the knee joint and fed a high fat diet containing 0.23% rice+0.018% AFR diet (OVX-OA-RAFR-L), (2) MIA injection into the knee joint and fed a high fat diet containing 0.69% rice+0.053% AFR diet (OVX-OA-RAFR-H), (3) MIA injection into the knee joint and fed a high fat diet added 0.053% cellulose (OVX-OA), or (4) saline injection into the knee joint and fed a high fat diet containing 0.053% cellulose (OVX). Sham rats had the same diet as control and saline injection (Sham; normal-control). At the beginning of the 8th week, an articular injection of monoiodoacetate into the right knee was performed in all OVX groups except the normal-control group and the assigned diets were provided for an additional 3 weeks. Values are mean ± SD. ^a,b,c^ Values on the same row with different superscripts were significantly different among groups by Tukey test at *p* < 0.05.

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
