# Peer review of "Rice Porridge Containing Welsh Onion Root Water Extract Alleviates Osteoarthritis-Related Pain Behaviors, Glucose Levels, and Bone Metabolism in Osteoarthritis-Induced Ovariectomized Rats"

_nutrients, 2019, doi:10.3390/nu11071503_

Round 1
Reviewer 1 Report
In the present manuscript the authors study the anti-osteoarthritis
effects of a phyto-molecule (allium fistulosum root water extract) in menopausal rats. They use a MIA-induced osteoarthritis
model, which is pertinent in vivo model in rats. Menopausal symptoms of ovariectomized rats were validated. By convincing in vivo experiments (at biochemical, behavioral, molecular and histological levels), they demonstrate that allium fistulosum root water extract
alleviates the MIA-induced effects.
Method for serum 17b-estradiol level quantification is missing, and should be added.
Minor comments:
- avoid abreviations in the abstract
- explain any abreviation used: BMD, ALP, LBM...
- tables 2, 3 and 4: typing errors in titles (RARF instead of RAFR)
- table 2: typing error in superscript for uterine weight (should be "b" for RAFR-H, and "a" for normal-control)
Author Response
We appreciate a good comment for our paper. We sincerely revised the manuscripts according to each comment. We changed the manuscript in red text to be easily distinguished.
Method for serum 17b-estradiol level quantification is missing, and should be added.
: We added it in the method section.
Minor comments:
- avoid abreviations in the abstract
: We removed the abbreviations in the abstract except the name of the groups.
- explain any abreviation used: BMD, ALP, LBM...
: We gave the full name.
- tables 2, 3 and 4: typing errors in titles (RARF instead of RAFR)
: We changed them.
- table 2: typing error in superscript for uterine weight (should be "b" for RAFR-H, and "a" for normal-control)
: We changed them.
Reviewer 2 Report
Major comments:
Rice porridge without root extract. How do we know that the observed effects are of the root extract and not by the porridge? There is no control group fed by rice porridge alone.
General advice – Throughout the text, the authors describe the normal controls as such: a decrease, an increase, improved, etc. as compared with the OA-control or OVX-control. This sounds unreasonable, because all the OVX groups are compared with the normal-control (Sham). For example, the normal-control rats cannot show an improved cartilage morphology. The OVX/OA rather show worse morphology as compared with normal-controls, and the RAFR feeding improves it. Please go all over the text, as such description mistakes are quite common.
The designations of the groups are confusing. Because 4/5 groups are OVX rats. I suggest the following designations: sham or controls, OVX, OVX-OA, OVX-OA-RARF-L, OVX-OA-RARF-H. Please correct it throughout the text.
Other comments:
Line 18 - RAFR-L, RAFR-H - what L and H stand for? High and low, respectively? Please mention that estrogen-deficient rats undergo ovariectomy (OVX).
Line 19 – “40 energy% fat” – change into “40% fat energy”.
Line 299-301 – The cause and effect between pain and food intake is only suggestive.
Abbreviations of LBM and FM are missing, lean body mass and fat mass, respectively?
Fig. 3B – the x axis titles are partially hindered by the bars.
Fig. 4C – y axis - please use “OA-induced leg” instead of “right leg”.
Fig. 4D – It’s unclear from the curve, whether velocity of RAFR-fed mice is increased statistically as compared with OA-control.
Fig 5. – If possible, zymography and/or ELISA from the cartilage could show anti-inflammatory effects by RAFR.
Line 535 – MMP3 -> MMP-13
Author Response
We appreciate a good comment for our paper. We sincerely revised the manuscripts according to each comment. We changed the manuscript in red text to be easily distinguished.
Rice porridge without root extract. How do we know that the observed effects are of the root extract and not by the porridge? There is no control group fed by rice porridge alone.
: I understand your point. However, we studied porridge containing the root extract as a whole since Asians consume rice porridge containing different extracts as a meal. The nutrient composition of diet was equivalent among the groups. In other words, the contents of carbohydrate, protein and fat were equivalents. We measured the carbohydrate, protein and fat in the porridge and omitted some from the control diet.
General advice – Throughout the text, the authors describe the normal controls as such: a decrease, an increase, improved, etc. as compared with the OA-control or OVX-control. This sounds unreasonable, because all the OVX groups are compared with the normal-control (Sham). For example, the normal-control rats cannot show an improved cartilage morphology. The OVX/OA rather show worse morphology as compared with normal-controls, and the RAFR feeding improves it. Please go all over the text, as such description mistakes are quite common.
: I appreciate your concerns. We checked the manuscript thoroughly and made sure the correct explanation.
The designations of the groups are confusing. Because 4/5 groups are OVX rats. I suggest the following designations: sham or controls, OVX, OVX-OA, OVX-OA-RARF-L, OVX-OA-RARF-H. Please correct it throughout the text.
: We changed the group names as suggested throughout the manuscript.
Other comments:
Line 18 - RAFR-L, RAFR-H - what L and H stand for? High and low, respectively? Please mention that estrogen-deficient rats undergo ovariectomy (OVX).
: It was changed.
Line 19 – “40 energy% fat” – change into “40% fat energy”.
: It was changed.
Line 299-301 – The cause and effect between pain and food intake is only suggestive.
: We changed it.
Abbreviations of LBM and FM are missing, lean body mass and fat mass, respectively?
: We gave the full name of LBM and FM was removed.
Fig. 3B – the x axis titles are partially hindered by the bars.
: It was changed.
Fig. 4C – y axis - please use “OA-induced leg” instead of “right leg”.
: It was changed.
Fig. 4D – It’s unclear from the curve, whether velocity of RAFR-fed mice is increased statistically as compared with OA-control.
: We wrote the statistical differences in the figure 4D.
Fig 5. – If possible, zymography and/or ELISA from the cartilage could show anti-inflammatory effects by RAFR.
: The cartilage was very small amounts and it was used for measuring mRNA expression. We do not have samples for protein assays.
Line 535 – MMP3 -> MMP-13
: It is changed.